# Overtreatment in Restorative Dentistry: Decision Making by Last-Year Dental Students

**DOI:** 10.3390/ijerph182312585

**Published:** 2021-11-29

**Authors:** Triana Moreno, José Luis Sanz, María Melo, Carmen Llena

**Affiliations:** Departament of Stomatology, Universitat de València, 46010 València, Spain; gaalle8bloo@hotmail.com (T.M.); jose.l.sanz@uv.es (J.L.S.); llena@uv.es (C.L.)

**Keywords:** overtreatment, overdiagnosis, restorative dentistry, dental caries

## Abstract

To evaluate the tendency for overdiagnosis and overtreatment upon different clinical situations among last-year students from the degree in dentistry from Valencia University (Spain) during the 2018–2019 course. A questionnaire consisting of 10 case exemplifications examining the diagnosis, treatment, and clinician’s attitude towards different common clinical situations regarding restorative dentistry was designed. Fifty-two students were surveyed, from whom 42 completed the questionnaire (80.77%). Data were analyzed descriptively. A total of 58.8% of the students correctly identified an early carious lesion in the occlusal surface of a molar, while 63.2% would perform unnecessary complementary tests for its diagnosis. The treatment for carious lesions in different evolutive phases with a vital pulp was correct between 51.2 and 92.7% of the cases. The treatment for irreversible pulp pathology and the restoration of the tooth with root canal treatment were adequately selected in 56.1% and 78.3% of the cases, respectively. For the repair of a faulty restoration, an overtreatment was proposed by 87.8% of the students. A tendency to perform unnecessary complementary tests for caries diagnosis was observed. Treatment caries proposals were in accordance with available evidence in the majority of the cases. Students tended to overtreat defective restorations and would perform unnecessary treatments in medically compromised patients.

## 1. Introduction

Overtreatment is defined as the application of unnecessary treatments or treatments whose efficacy has not been demonstrated and there is limited or no evidence that supports their beneficial effect on the patients’ health [1]. Likewise, the definition encompasses any medical procedure that is technically and clinically inadequate due to the existence of other exceeding options, or any treatment whose volume and/or price are inappropriate [2].

Overdiagnosis and overtreatment within the field of dentistry constitute fraudulent and unethical conduct, which poses a situation of conflict between the patients’, the service providers’, and funders’ interests. Occasionally, the lack of standardized protocols results in variations in treatment decisions which, although technically adequate, may not be as so from the patients’ health point of view depending on its individual conditions. Thus, the communication with the patients in understandable terms, and the consensual search for the best therapeutic option for each case is essential for an ethical dental practice based on patients’ informed consent [3,4].

There will be a risk of overtreatment and infringement of the ethical principles of the dental practice when the economic interest comes first than the patients’ needs [4,5]. The same occurs when a diagnosis is established as conditions which will never cause symptoms or pain, commonly referred to as overdiagnosis. In fact, in some cases, for the treatment of these conditions, the patient may even experience adverse effects instead of benefits [6].

In restorative dentistry, overtreatment appears as a result of an overdiagnosis and/or proposal of unnecessary invasive treatment plans. This attitude is usually accompanied by a limited communication with the patient regarding the process of his/her problem and the necessary measures to limit or revert it, leading the patient to a cycle of restoration and re-restoration [7]. The same occurs when the complete removal and replacement of defective restorations is proposed, or when more complex and/or aggressive techniques are primarily recommended [8,9].

Dental caries is a highly prevalent disease with an important impact in oral health. Restorative treatments represent the highest workload among dental clinics, as well as a high cost, not only economic, but also in biological terms and in the quality of life of the patients. Despite the extension of caries-prevention programs, in the younger population, the prevalence of the disease and the need for its treatment maintains elevated [10].

The consideration of dental caries as an ecological alteration in the dental biofilm has fixed new strategies for its control, looking to alter its development and growth, as well as modifying the progress of the dissolution of the apatite component within the tooth’s structure, by applying basic measures such as the control of the sugar intake and the mechanical removal of biofilm [11].

Dental schools have favored the disciplines directed at learning and applying increasingly sophisticated restorative techniques; to the detriment of the knowledge of the natural progression of the disease, the application of health education measures, and the consideration of individual and community risk factors [12].

Currently, the approach for caries disease has to be developed from the knowledge and control of risk factors, defined by the International Caries Classification and Management System (ICCM) [13]. The ICCM provides a structural diagnostic and treatment process for caries which is centered around the patient, considers the evolutive state and activity of the lesion, and is aimed at the inactivation and control of the disease process, the preservation of dental tissues and, ultimately, the preservation of the tooth for the longest possible [14,15]. Other approaches are to be considered as overtreatment and thus are ethically unacceptable [1,16].

In relation to this concern, it is of interest to evaluate the perspective among dental students regarding the diagnostic and therapeutic approach for caries disease and its consequences on the life cycle of the teeth and the patients’ health. To the authors’ knowledge, there is a limited number of studies assessing the tendency for overdiagnosis and overtreatment [1,17,18]. The present study aims to evaluate overdiagnosis and overtreatment upon different clinical situations among last-year dental students.

## 2. Materials and Methods

The present study was previously approved by the ethics committee from de Valencia University (Spain) (ref. 1235331).

An original questionnaire was designed, based on previous surveys performed in different countries, which included diagnostic and treatment alternatives for varying clinical situations related to restorative dentistry [1,7,19]. 

Different scenarios were included in the form of clinical and/or radiographic images that allowed a clear and rapid identification of the situation, which was to be evaluated, along with a brief description of the clinical case.

The following clinical situations were included: four cases illustrating carious lesions in different evolutive phases, two cases exhibiting pulp damage, one case of a dental fracture, one case involving consumptive processes (i.e., structural loss by attrition, erosion, etc.) and multiple tooth losses in a medically compromised patient, one case of a defective restoration, and one case of tooth discoloration. Each scenario consisted of at least one question regarding the diagnosis or the treatment plan for the illustrated case. Various answer formats were included along the questionnaire: short answer, multiple choice, and ranking answer (by ordering a series of options from highest to lowest in terms of their suitability).

To establish the correct answers for the cases, as well as their adequate evaluation, scientific evidence of the highest quality possible was searched. Whenever possible, systematic reviews of randomized controlled trials (evidence level 1a) were used. Alternatively, the article with the highest evidence available was selected. The clinical situations and the references of the articles used to establish the answers and their adequate evaluation are presented in Table 1 [1,3,10,13,14,15,16,20,21,22,23,24,25,26,27,28,29,30,31,32].

Once established, the questionnaire was shown to 5 dental students and 5 dentists with more than 10 years of experience, to make sure that the clinical cases were clear and the diagnostic and therapeutic options were comprehensible. Their opinions were evaluated, and any relevant drafting modifications they suggested were applied to design the final structure of the questionnaire.

The questionnaire was distributed to all last-year students from the degree in dentistry from Valencia University (Spain) during the 2018–2019 course (*n* = 52), via an electronic link from the university’s virtual platform. To do so, the questionnaire was introduced in LimeSurvery platform. The platform assigned a number for each survey according to the date of its completion, without registering any information from the surveyed participant; thereby producing anonymized answers.

Any questionnaire that was not completed in its totality was excluded from the analysis. A score of 1 or 0 was assigned to the answers depending on their adequacy, according to the previously established criteria. A descriptive analysis of the data was performed using the SPSS 20.0 (SPSS Inc., Chicago, IL, USA).

## 3. Results

A total of 52 students were surveyed, from whom 42 answered the totality of the questionnaire (80.77%). The mean age of the participants was 25.22 ± 5.44 years old; 70.7% of the participants were female.

### 3.1. Clinical Scenario #1

The question from scenario 1a was open-ended; participants were asked to provide a diagnosis of a lesion in tooth 3.6. A total of 41.5% of the surveyed students considered that the image presented a retentive pit/fissure or stained pit/fissure; 39% of them answered that the image exhibited a carious lesion. The answers “healthy”, “early carious lesion”, and “chronic carious lesion” were less frequent: 2.4%, 9.8%, and 7.3%, respectively. For this question, the correct answers were: “retentive pit/fissure or stained pit/fissure” and “chronic carious lesion”. “Healthy” and “carious lesion” were considered as incorrect. Thereby, 48.8% of the surveyed students answered correctly (Figure 1).

For scenario 1b, students were required to order different complementary diagnostic tests from more to less adequate for the diagnosis of the presented case. The most prevalent answer, “bitewing X-ray” (51.2%), “periapical X-ray” (12.2%), and “no complementary diagnostic test is required” (36.6%). The latter was the option considered as correct (Figure 1).

In a similar manner, the surveyed students were asked to order different therapeutic options in scenario 1c. The option “oral health instructions and 6-month follow-up visit” was selected by 51.2% of the participants. This answer was considered as correct Additionally, 9.8% of the students would perform a “pit and fissure sealing”, 7.3% would opt for a “2-year follow-up visit”, and 2–4% would carry out a resin-based obturation/filling (Figure 1).

### 3.2. Clinical Scenario #2

In this scenario, students were required to order different therapeutic alternatives from more to less adequate as a treatment for the presented case. The majority of the surveyed students (58.5%) selected “pulp capping” as their first choice. From more to less prevalent, the remaining treatment options were: “maturogenesis” (29.3%), “apexification” (9.8%), and “conventional root canal treatment” (2.4%). Only “maturogenesis” and “pulp capping” were considered as correct, meaning that 87.8% of the surveyed students provided a correct answer (Figure 1).

### 3.3. Clinical Scenario #3

The surveyed students were asked to select the most adequate treatment option for tooth 4.6. A total of 70.7% of the participants agreed that the most adequate treatment would be to give oral hygiene and dietary instructions, apply a remineralizing agent, and perform a 1-year follow-up clinical and radiographic examination. Additionally, 19.5% of the surveyed students would perform a “restoration with resin composite”, and 4.9% selected “pit and fissure sealing” or “6-month follow-up visits” (Figure 1).

### 3.4. Clinical Scenario #4

With regards to this clinical scenario, the majority of surveyed students (92.7%) selected, as a first treatment option, “direct pulp capping”. Among those who selected “direct pulp capping”, 68.4% would then place a “temporary restoration”, and 31.6% a “definitive restoration”. Additionally, 2.4% would perform a “root canal treatment” and then place a “definitive restoration”, 4.9% would carry out a “root canal treatment” and then place a “definitive restoration and a crown”. The alternative “direct pulp capping and definitive restoration” was considered as correct (Figure 1*).*

### 3.5. Clinical Scenario #5

In this scenario, the participants were asked to select the most adequate treatment option for the treatment of the carious lesions present in teeth 1.3, 1.2, 1.1, 2.1, 2.2, and 2.3. The most prevalent answer for teeth 1.3, 1.1, 2.1, and 2.3 (94.2%) was “remineralization” (clinical scenario 5a), which was the correct answer. For tooth 1.2 (clinical scenario 5b), the most prevalent answer was “root canal treatment” (51.2%), followed by “restoration with resin composite” (22%). For tooth 2.2 (clinical scenario 5c), the most popular option was “restoration with resin composite” (85.4%), followed by “remineralization” (22%). The correct answer for teeth 1.2 and 2.2 was “restoration with resin composite” (Figure 1). 

### 3.6. Clinical Scenario #6

The first question (clinical scenario 6a) consisted of two options: “root canal treatment” and “extraction”, and students were required to select which of the proposed options would be most adequate for the proposed case. In this case, the correct alternative was “root canal treatment”, which was selected by 56.1% of the students.

The second question (clinical scenario 6b) was asked to those who selected “root canal treatment” as the treatment for the proposed case, and required them to order different alternatives for the restoration after root canal treatment. The “direct restoration with resin composite” was selected by 78.3% of the students, which was the correct option. Additionally, 13% of the students would additionally place a post and a crown. The least repeated restoration among the surveyed students was “incrustation” (8.7%).

The third question (clinical scenario 6c) should only be answered by those who selected “extraction” as the treatment for the proposed case, and required them to order different alternatives for the tooth replacement. The options “deferred implant” or “fixed prosthesis” were the most prevalent (44.4%). To a lesser extent (11.2%), students opted for the placement of an “immediate implant”. The placement of a “deferred implant” was considered as correct (Figure 1).

### 3.7. Clinical Scenario #7

In the seventh clinical scenario, students were required to order different therapeutic alternatives for the proposed case. A total of 48.8% of the surveyed students agreed that the most adequate option would be to “explain the different treatment alternatives to replace the tooth after extraction, with their strengths and limitations”, which was considered as the correct answer. An additional 24.4% selected the option “immediate implant and Michigan splint”, 22% would place a “removable prosthesis”, and 2.4% selected “deferred implant and Michigan splint” or “fixed prosthesis” (Figure 1).

### 3.8. Clinical Scenario #8

This scenario required, again, that the students ordered different therapeutic alternatives for the proposed case. The most prevalent first option was “restoration of the attritions” (31.7%), followed by “no restorative nor prosthetic treatment is required” (26.8%), “teeth-supported rehabilitation” (22%), “removable prosthesis” (14.6%), and implant-supported rehabilitation” (4.9%). The correct option was “no restorative nor prosthetic treatment is required” (Figure 1).

### 3.9. Clinical Scenario #9

Similar to clinical scenarios 7 and 8, students were required to order different treatment alternatives for the proposed case. A total of 43.9% of the students would “replace the restoration in its integrity with a new direct restoration”, 39% would “replace the restoration with an incrustation”, and 12.2% would “replace the restoration partially and restore the anatomy of the tooth”. This last option was considered as correct (Figure 1).

### 3.10. Clinical Scenario #10

The last clinical scenario was divided in two parts. For clinical scenario 10a, the majority of the surveyed students selected an “internal bleaching” (90.2%) as the most adequate treatment option, which was considered as correct. A small percentage of students opted for a resin composite veneer (7.3%) or a ceramic veneer (2.4%).

For the second part (clinical scenario 10b), students were asked to select their attitude towards the patient asking for an extraction of the affected tooth and the placement of an implant. The majority of the students (82.9%) would not comply with the patient’s demands and would explain the more conservative alternatives available. For this case, this is the only correct therapeutic attitude (Figure 1).

## 4. Discussion

### 4.1. Analysis of the Diagnostic Criteria

The results from the present study confirmed that approximately half of the last-year students from the degree in dentistry proposed a correct diagnosis for the first clinical scenario, following the criteria established by The International Caries Consensus Collaboration or ICCC [33], in which the extension and activity of the lesion are considered. The question was open-ended, and a high percentage of participants answered “caries”, without providing more details regarding the activity of the lesion.

As advised by the ICCC, the diagnosis of caries, nowadays, has to be based on the visual clinical exploration and the evaluation of individual risk of caries development [34]. Diagnostic tests will act as supporting material for the determination of the extension of the lesion and to control its progression over time. The majority of students selected, as a complementary test, performing a bitewing X-ray to confirm de diagnosis. This could be considered as an indication for “overdiagnosis”, as this test, in the case of an inactive carious lesion limited to the superficial enamel on the occlusal surface, does not provide any additional help to the diagnosis [13]. This tendency to use complementary tests could be explained by the clinical inexperience of the students and by the excessive trend to support diagnosis on complementary tests (Figure 1).

### 4.2. Analysis of the Therapeutic Criteria

Currently, there is a great tendency towards the restoration of lesions limited to the enamel, although the scientific evidence supports other non-invasive alternatives for the control of these lesions [35]. According to these criteria, an inactive carious lesion categorized as ICDAS II (clinical scenario 1c) should not receive any restorative treatment, but only require oral health instructions and control of individual risk factors, together with follow-up control visits [35]. Interestingly, the percentage of students who correctly answered the therapeutic approach in scenario 1c (51.2%) was higher than those who correctly answered the diagnosis in scenario 1a (48.8%). This difference may have been due to the fact that students who misdiagnosed scenario 1a as “healthy” or “early carious lesion” may have selected “oral health instructions and 6-month follow-up visit” as a therapeutic approach. However, a little more than half of the students selected this non-invasive therapeutic alternative. It should be highlighted that up to 41.7% of the students would place a resin composite restoration, which in this case is considered as an “overtreatment”. Available systematic reviews among the literature regarding the treatment for early carious lesions found a significative proportion of dentists who would propose restorative treatments upon carious lesions for which minimally invasive technique are indicated [30,32]. Thus, it is necessary to orient dental students into a less invasive approach, taking into account the natural progression of the disease, and treating lesions according to their extension and carioactivity. Because the influence of the type of undergraduate formation influences the postgraduate therapeutic attitude [36].

The majority of students coincided with the available protocols for approaching active carious lesions limited to enamel (clinical scenario 5a). However, when lesions extended to dentin (clinical scenarios 5b and c), the therapeutic alternatives were more discordant. Specifically, for the treatment of tooth 1.2, in which the extension of the carious lesion surpassed the external third of the dentin, 51.2% of the students proposed a root canal treatment, which is considered as an overtreatment in this case [35].

Upon carious lesions with a greater extension, the therapeutic approach becomes varied depending on the extent of the lesion. When the pulp is reversibly affected, or an accidental pulp exposure is produced when removing affected tissue from deep carious lesions, the preservation of pulp vitality is preferable [31,37]. A direct pulp capping is indicated in cases where an asymptomatic pulp exposure with a controllable bleeding is produced, and the tooth is restorable. Pulpotomy is reserved for pulp exposures of greater size (>1 mm) and whose bleeding can be controlled within a period of 1–2 min [21]. The results from our study show that the surveyed students majorly coincide with the current recommendations for the treatment of the vital pulp. The participants would perform a pulp capping on clinical scenarios 2 and 4. However, the majority of them would then place a temporary restoration (i.e., two-stage selective caries removal), while available evidence supports one-stage selective removal procedure [20,38].

When the inflammation of the pulp is accompanied by periapical pathology, a more invasive operatory approach is required. In the case of suppurative chronic periapical periodontitis the ideal approach would be root canal treatment [21]. This situation was presented in clinical scenario 6a, where 56.1% of the surveyed students opted for root canal treatment. Those who selected this treatment option then would restore the tooth with a resin composite (clinical scenario 6b), which was the answer considered as correct. For the treatment of small or medium-sized cavities, where there is enough dental structure to retain a restoration, it would be the most suitable option [29]. Manocci et al. concluded that the outcome of a direct restoration at 3 years of follow-up was the same as the obtained by placing a crown [39].

Approximately 44% of the surveyed students answered that an extraction would be the most suitable option for clinical scenario 6a, which would imply a more aggressive and expensive treatment for the patient if a prosthetic replacement of the tooth with an implant is planned (as answered by 44,4% of the participants in clinical scenario 6c). The investigators proposed the placement of a deferred implant as the most suitable option for the replacement of the tooth if opting for its extraction, since the replacement of a single tooth implies less removal of tooth structure than a fixed prosthesis. Currently, deciding between an immediate or deferred implant is a controversial matter, since there is a great variability among studies in the field, which should be interpreted with caution [40]. The investigators opted for the deferred implant, since the case presented an infectious process which involved the osseous tissue, which could hinder the prognosis of an immediate implant.

The replacement or repair of defective restorations are common techniques among daily dental practice [24]. In a clinical study by Moncada G. et al., the effectiveness of different therapeutic approaches for the treatment of defective restorations was evaluated for 3 years. The authors observed that, by replacing the defective restoration in its integrity, no greater durability was achieved when compared to a more conservative approach, i.e., repairing the defective restoration [41]. Only 12.2% of the surveyed students would have opted for the repair of the defective restoration presented in clinical scenario 9. The rest of the participants selected the option involving the complete replacement of the defective restoration or the placement of an incrustation. Both options could be performed and can be adequate from a technical point of view, but would not be as conservative [24].

In clinical scenario 8, which presented the case of a patient undergoing head and neck irradiation who referred trismus due to the ankylosis of the temporo-mandibular joint. The majority of students proposed various options of rehabilitator treatments. Only 26.8% of the surveyed students proposed no prosthetic treatment, but only maintenance and preventive measures. Since the patient did not present any clinical symptoms, the most suitable option is the application of basic measures and the resolution of any possible dental complications that could present in the most conservative way possible [23].

A common case among restorative dentistry is the treatment of dental discoloration after root canal treatment. Clinical scenario 10a presented this situation, which more than 90% of the students resolved by dental bleaching as a first option. In a systematic review of in vitro studies, dental bleaching is described as an adequate approach for these situations, being the most conservative approach available [26].

### 4.3. Analysis of the Professional Attitude

In clinical scenario 7, in which tooth extraction was presented as the only option for treatment, only 48.4% of the surveyed students would have opted for discussing other treatment options with the patient; which would have been the most adequate approach [3].

The objective of clinical scenario 10b was to assess the attitude of the professional upon a patient who demanded an inadequate treatment for his/her needs, which could not only lack any benefit for him/her, but even be harmful in terms of functionality and aesthetics. In this case, 82.9% of the participants considered that the best attitude towards this situation was more information about alternative treatments and patient informed consent, explaining the reasons and offering diverse alternatives to solve his/her problem/s. The unnecessary overtreatment demanded by the patient would be more destructive, non-beneficial for his/her health and contrary to the ethical principles of our profession [1,27,42].

In light of the results obtained, it would be desirable to have unified protocols, with standardized criteria for undergraduate students. This, together with the approach to clinical practice from an integrated perspective, rather than by subject, could improve student training towards a more conservative approach, which would lead to a more rational approach in the use of diagnostic procedures and in the application of therapeutic procedures.

## 5. Conclusions

In the present study, it was observed that last-year students from the degree in dentistry tended to perform unnecessary complementary diagnostic tests upon occlusal carious lesions, while their treatment planning upon carious lesions of different extension was in accordance with the available evidence in the majority of cases. At the same time, they showed difficulty when facing the treatment of a defective restoration and tended to perform unnecessary treatments in patients with medical compromise. Furthermore, the overtreatment in restorative dentistry would be non-beneficial for patient health and contrary to the ethical principles of our medical profession.

## Figures and Tables

**Figure 1 ijerph-18-12585-f001:**
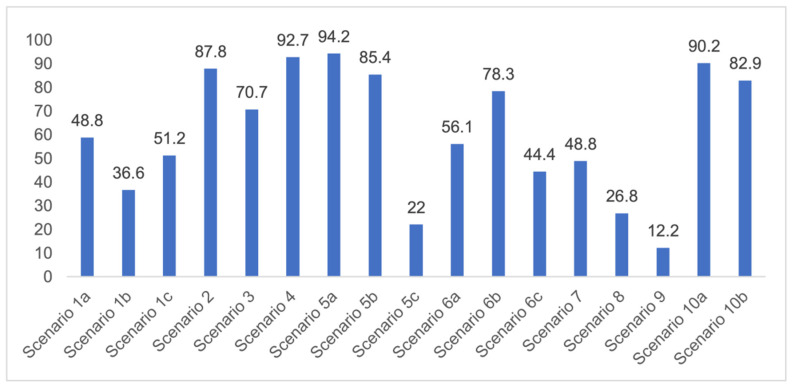
The percentage of correct questions answered by the surveyed students (*n* = 42) for each of the clinical scenarios.

**Table 1 ijerph-18-12585-t001:** Clinical scenarios and references from the scientific evidence consulted for each case.

Clinical Scenario	Description	Images	Question		Answers	Reference
1	A 21-year-old healthy patient visits the dental clinic for a check-up. The patient does not present any symptoms nor other oral clinical signs, except visible plaque in all quadrants, which extends towards the gingival third of various teeth.	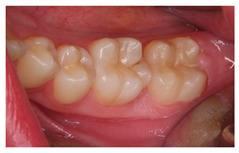	Regarding the distal pit of tooth 3.6 What is your clinical diagnosis? Which complementary diagnostic tests would you perform to confirm your diagnosis for tooth 3.6? Order the answers from more to less relevant for the diagnosis of this case.		CBCT Periapical x-ray Ortopantomography Bitewing x-ray No complementary diagnostic test is required	[13,14] [30,31]
			Which treatment alternative would you select for this case? Order the answers from more to less relevant for the treatment of this case.		Restoration with resin composite OHI and 6-month follow-up visit Pit and fissure sealing 2-year follow up visit, the patient is healthy OHI and restoration with resin composite	
2	An 11-year-old patient.Upon clinical examination, multiple carious lesions are observed in various deciduous teeth and permanent first molars. In tooth 4.6, the patient reported sensitivity to cold stimuli, without episodes of intense pain nor inflammation. Additionally, there is an absence of pain to percussion and physiological probing depth.	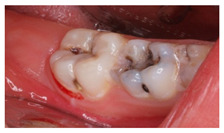 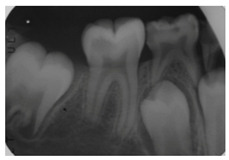	The diagnosis was reversible pulpitis in an immature tooth. Which treatment alternative would you select for this case? Order the answers from more to less adequate for the treatment of this case.		Extraction Pulp capping Apexification Maturogenesisis Conventional root canal treatment	[10,13,14,30,32]
3	A 29-year-old patient visits the dental clinic to evaluate his/her general oral state. A complete clinical examination is carried out and no pathological signs were observed. Bitewing x-rays are performed, and various radiolucent lesions limited to the enamel were observed on the right bitewing x-ray.	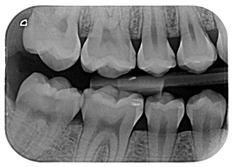	Select the most adequate treatment alternative for tooth 4.6 from the answers provided.		Restoration with resin composite 6-month follow-up visits OHI, dietary instructions, remineralizing agent, and 1-year follow-up visit and x-ray. Only dietary instructions Fissure sealing	[13,14]
4	16-year-old patient. Upon removal of a carious lesion, an accidental pulp exposure in 4.6 is produced, as shown in the image. Clinically, the patient reported no symptoms before the treatment procedure.	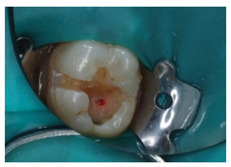	Which treatment alternative would you select for this case? Order the answers from more to less relevant for the treatment of this case.		DPC, definitive restoration and crown DPC and definitive restoration DPC and temporary restoration Root canal treatment and definitive restoration Root canal treatment, definitive restoration and crown	[13,15,16,20,21,30]
5	In the image, various carious lesions in different evolutive states are presented 15 year-old patient.	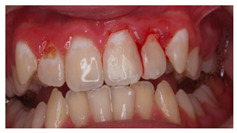	Which treatment alternatives would you select for this case? Select, individually, the treatment for teeth 1.3, 1.2, 1.1, 2.1, 2.2 y 2.3 from the given answers.		Remineralization Restoration with composite resin Root canal treatment No treatment	[11,13,14,32]
6	A 40-year-old patient visits the dental clinic reporting repeated episodes of inflammation and pain in the posterior area of third quadrant. Clinically, the patient currently presents a suppurative and inflamed area near tooth 3.6, referring only occasional mild pain. After the clinical and radiographic examination, a diagnosis of chronic apical periodontitis with an episode of re-exacerbation is reached.	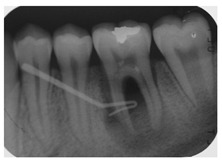	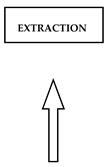	If you have decided extraction. Which treatment alternative would you select to replace tooth 3.6?	Removable partial prosthesis Fixed prosthesis Immediate implant Deferred implant	[22,28]
			Which treatment alternative would you select for this case?			[21]
			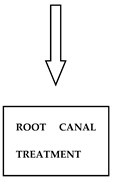	If you have decided root canal treatment. Which treatment alternative would you select to restore the crown of tooth 3.6?	Direct restoration with resin composite Incrustation Direct restoration and post Direct restoration, post and crown	[29]
7	A 58-year-old patient visits de dental clinic referring pain when chewing in the third quadrant, which occasionally irradiates towards the ear. Upon clinical examination, a visible fracture is observed in tooth 3.6, with a mild mobility of the mesial fragment. The radiographic examination reveals a line of fracture from the crown to the mesial root. The tooth is diagnosed with a vertical fracture and its extraction is recommended.	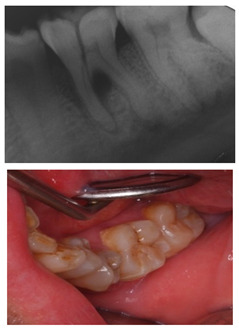	Which treatment alternative would you select for this case? Order the answers from more to less relevant for the treatment of this case.		Explain the different treatment alternatives to replace the tooth after extraction Immediate implant and Michigan splint Deferred implant and Michigan splint Fixed prosthesis Removable prosthesis	[3,31]
8	A 59-year-old patient underwent surgery and head and neck radiotherapy for a squamous cell carcinoma on the tongue 10 years ago. After radiotherapy, there is a limitation of the oral opening (1 cm) due to a TMJ ankylosis. He refers frequent episodes of mouth ulcers which heal slowly, xerostomy and oral candidiasis.	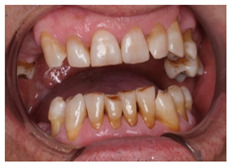	Which treatment alternative would you select for this case? Order the answers from more to less relevant for the treatment of this case.		Restoration of the attritions Implant-supported rehabilitation Tooth-supported rehabilitation Removable prosthesis No restorative nor prosthetic treatment is required	[23]
9	A 16-year-old patient visits de dental clinic and upon clinical examination, you observe the restoration shown in the image on tooth 2.6. It was placed 3 years ago to treat a carious lesion which developed over a molar-incisor-hypomineralization defect. The restoration shows clear signs of microfiltration and the mesial-buccal susp is fractured as a result of the mineralization defect.	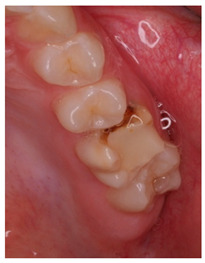	Regarding the resin composite restoration shown in the image, how would you proceed?		Replace the restoration in its integrity with a new direct restoration Replace the restoration with a crown Replace the restoration with an indirect restoration Replace the restoration partially and restore the anatomy of the tooth	[24,25,31]
10	A 58-year-old patient with history of trauma on the anterior region visits the dental clinic. The patient reported a traumatic lesion 20 years ago, developing a phlegmon which later developed into a sinus tract originating from tooth 2.1. A root canal treatment was performed on tooth 2.1 and has been darkening since then, in a progressive manner. The patient requests a treatment for the change in color. The tooth does not present mobility nor pathological probing depth. The radiographic examination reveals a correct root canal treatment without pathological signs.	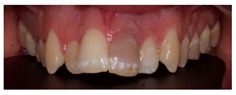	Which treatment alternative would you select for this case? Order the answers from more to less relevant for the treatment of this case.		Crown Internal bleaching Ceramic veneer Resin composite veneer Extraction, orthodontic treatment, and prosthetic rehabilitation	[26]
			If the patient requests the extraction of tooth 2.1 and placement of an implant, how would you proceed?		I would perform the extraction, explaining the limitations of losing a central incisor I would not perform the extraction, explaining the limitations of losing a central incisor I would not perform the extraction, explaining the more conservative alternatives available	[1,27]

CBCT: cone-beam computed tomography; OHI: oral health instructions; DPC: direct pulp capping, TMJ: Temporo mandibular joint.

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
