# Peer review of "Overtreatment in Restorative Dentistry: Decision Making by Last-Year Dental Students"

_ijerph, 2021, doi:10.3390/ijerph182312585_

Round 1

Reviewer 1 Report

This article did assess the decision making and tendency of over treatment by last-year dental students.

This is an interesting article. The methodology and the general setup is appropriate. However, the article is missing out of an opportunity by not introducing additional groups for example: dentist with 1, 5 and 10 year experience.

Discussion: Please discuss in light of the gained knowledge, which measures would be beneficial to target this issue (overtreatment in dentistry)

Author Response

This article did assess the decision making and tendency of over treatment by last-year dental students.

This is an interesting article. The methodology and the general setup is appropriate. However, the article is missing out of an opportunity by not introducing additional groups for example: dentist with 1, 5 and 10 year experience.

Thank you very much for your comment. It would certainly have been interesting to be able to include data relating to professionals with years of experience. We initially planned to do so, but the number of responses from professionals was very limited and consequently it was not representative.

Discussion: Please discuss in light of the gained knowledge, which measures would be beneficial to target this issue (overtreatment in dentistry)

The following paragraph has been added and marked in red in the text.

In light of the results obtained, it would be desirable to have unified protocols, with standardized criteria for undergraduate students. This, together with the approach to clinical practice from an integrated perspective, rather than by subject, could improve student training towards a more conservative approach, which would lead to a more rational approach in the use of diagnostic procedures and in the application of therapeutic procedures.

Reviewer 2 Report

I appreciate the authors trying to measure the diagnostic and therapeutic decisions made by dental students. However, there are serious concerns related to the sample size and methodology of the study.

The standardization for choosing the correct answers (Appropriate diagnosis or treatment) is not explained and the method used is questionable.

Validation of the questionnaire is not adequately explained.

Author Response

I appreciate the authors trying to measure the diagnostic and therapeutic decisions made by dental students. However, there are serious concerns related to the sample size and methodology of the study.

Thank you very much for your comments.

Regarding the sample size, all students in the final year of the degree in dentistry (52 students) were invited to participate, and the questionnaires that were partially completed were excluded, leaving a total of 42 questionnaires for analysis.

We could have planned a multicenter study, which would have allowed us to have a larger sample size and to evaluate differences depending on differential factors in the teaching of the different centres, but this was not our objective. That is why the sample size is indeed limited.

The standardization for choosing the correct answers (Appropriate diagnosis or treatment) is not explained and the method used is questionable.

The decision as to which was the correct answer was established on the basis of the available evidence. Whenever possible, systematic reviews were chosen to establish the diagnostic or therapeutic criteria (see references 15,26,34 and 36). When this was not possible, consensus documents were used (see references 11,13,14, and 33). When this was also not possible, references with the best possible evidence were used. The references used for each scenario can be seen in table 1. References relating to ethical aspects of the profession were also used. Some references that were missing in table 1 have been introduced (marked in red and the corresponding paragraph in the text has been modified).

Validation of the questionnaire is not adequately explained.

The respective paragraph has been modified and marked in red in the text.

Once the scenarios had been developed, in order to assess the validity of the content, 5 students and 5 professors from the degree in dentistry -one of them a professor of bioethics with more than 10 years of experience- were asked to solve the scenarios. The students were asked to answer  each of the scenarios anonymously and to make any comments they considered appropriate regarding the language and clarity of the questions and answers. With the 5 teachers, a working session was held in which they were asked to answer each of the scenarios, taking into account the diagnostic criteria and treatment they would propose, while taking into account the ethical considerations raised in this study, overtreatment or overdiagnosis (unnecessary diagnostic tests or treatments). In the responses where there was no consensus among the 5 teachers, the alternatives were discussed, and they were asked to review the scientific literature used by the authors of the scenarios until consensus was reached.

Reviewer 3 Report

MDPI/INTERNATIONAL JOURNAL OF ENVIROMENTAL RESEARCH AND PUBLIC HEALTH

            Rating the manuscript: Overtreatment in restorative dentistry. Decision making by last-year dental students.

  • Minor criticisms of the manuscript

ABSTRACT:

15 The data were analyzed descriptively instead data were analyzed in a descriptive manner

22 Treatment caries proposals instead therapeutic caries propositions

24 Medical compromise situation /patient instead in situations where a medical compromise

INTRODUCTION:

40 Ethical dental practice based on patient informed consent instead clinical practice based on ethical principles

48 Overdiagnosis and/or proposal for unnecessary invasive treatment plan instead

excessive diagnosis and/or when invasive treatment plan is posed.

60,61 The consideration…has fixed new strategies instead has posed..

MATERIALS & METHODS:

95 Medical compromise patient instead oncologic patient

Table 1

Clinical scenario# 2= 11-year-old patient- 46 which is correct diagnosis? maturogenesisis instead apexogenesis .

Clinical scenario# 3= 29-year-old patient-46 which is correct diagnosis ?

Clinical scenario # 4=16-year-old patient- which tooth?

Clinical scenario # 5= which is the age of this  patient? Maybe more information about treatment methods selected individually for each tooth?

Clinical scenario# 6= 40 –year-old patient. Which treatment alternative and why? Conservative treatment or extraction..

Clinical scenario# 8= 59-year-old patient –oncological or medical compromise patient with trismus ( 1 cm aperture) is not very clear why only the answer  " no restorative treatment nor prosthetic treatment is required" is accepted as correct.

Clinical scenario # 9= 16-year-old patient.The tooth 26, treated 3 years ago, more discussion about MIH defect

123 to make sure instead to ascertain

RESULTS:

3.1 Clinical scenario#1- A discussion about the reason why 48,8% of the surveyed students answer correctly to the diagnosis of the lesion ( 1a) and 51,2% answer correctly to the treatment option ( 1c).

DISCUSSION:

272 Minimally invasive technique instead less invasive treatments are indicated

275 To their extension and carioactivity instead to their extension and activity

348 More information about alternative treatments and patient informed consent instead to inform the patient that the treatment that he/she proposed cannot be performed

CONCLUSIONS:

Furthermore, the overtreatment in restorative dentistry would be non-beneficial for the patient health and contrary to the ethical principles of our medical profession.

Author Response

Rating the manuscript: Overtreatment in restorative dentistry. Decision making by last-year dental students.

  • Minor criticisms of the manuscript

All of the proposed minor changes have been implemented in the text (marked in red).

ABSTRACT:

15 The data were analyzed descriptively instead data were analyzed in a descriptive manner

22 Treatment caries proposals instead therapeutic caries propositions

24 Medical compromise situation /patient instead in situations where a medical compromise

INTRODUCTION

40 Ethical dental practice based on patient informed consent instead clinical practice based on ethical principles

48 Overdiagnosis and/or proposal for unnecessary invasive treatment plan instead

excessive diagnosis and/or when invasive treatment plan is posed.

60,61 The consideration…has fixed new strategies instead has posed..

MATERIALS & METHODS:

95 Medical compromise patient instead oncologic patient

Table 1

Clinical scenario# 2= 11-year-old patient- 46 which is correct diagnosis?  Reversible pulpitis in an immature tooth Maturogenesisisinstead apexogenesis.

Clinical scenario# 3= 29-year-old patient-46 which is correct diagnosis? Interproximal caries limited to enamel

Clinical scenario # 4=16-year-old patient- which tooth? 4.6

Clinical scenario # 5= which is the age of this patient? 15 years old

Maybe more information about treatment methods selected individually for each tooth?

A general treatment method description was included in order to avoid imposition of the use of a specific technique/material. Additionally, the mentioned methods are standardized treatment methods and are well-known among dental students and clinicians.

Clinical scenario# 6= 40 –year-old patient. Which treatment alternative and why? Conservative treatment or extraction.

Treatment alternatives were not in the table. Alternatives have been added and completed, depending on the treatment decision.

Clinical scenario# 8= 59-year-old patient –oncological or medical compromise patient with trismus ( 1 cm aperture) is not very clear why only the answer  " no restorative treatment nor prosthetic treatment is required" is accepted as correct.

This option was considered to be correct because the patient was irradiated in the head and neck and presented multiple root remnants that currently do not cause any acute infectious process and attritions. Extraction of these teeth could cause osteoradionecrosis. The patient does not report any dental sensitivity, or any problems derived from the attritions present. His main problems are those derived from hyposalivation and the side effects of radiotherapy which caused capsular fibrosis of the TMJ. For this reason, no interventional treatment is proposed (see reference 23).

Clinical scenario # 9= 16-year-old patient. The tooth 26, treated 3 years ago, more discussion about MIH defect

Additional information regarding the MIH defect has been added in clinical scenario #9.

123 to make sure instead to ascertain

RESULTS:

3.1 Clinical scenario#1- A discussion about the reason why 48,8% of the surveyed students answer correctly to the diagnosis of the lesion ( 1a) and 51,2% answer correctly to the treatment option ( 1c).

A comment has been added in the discussion section: “Interestingly, the percentage of students who correctly answered the therapeutic approach in scenario 1c (51.2%) was higher than those who correctly answered the diagnosis in scenario 1a (48.8%). This difference may have been due to the fact that students who misdiagnosed scenario 1a as “healthy” or “early carious lesion” may have selected “oral health instructions and 6-month follow-up visit” as a therapeutic approach.”

DISCUSSION:

272 Minimally invasive technique instead less invasive treatments are indicated

275 To their extension and carioactivity instead to their extension and activity

348 More information about alternative treatments and patient informed consent instead to inform the patient that the treatment that he/she proposed cannot be performed

CONCLUSIONS:

Furthermore, the overtreatment in restorative dentistry would be non-beneficial for the patient health and contrary to the ethical principles of our medical profession.

Reviewer 4 Report

The topic is interesting and has high potential considering that overtreatment represents a frequent situation in current practice. However, a high quality research must include a sample size calculation (in the current study I consider it is insufficient to provide significant results) and also include a more advanced statistic evaluation of results.

Author Response

The topic is interesting and has high potential considering that overtreatment represents a frequent situation in current practice. However, a high quality research must include a sample size calculation (in the current study I consider it is insufficient to provide significant results) and also include a more advanced statistic evaluation of results.

Thank you very much for your comments. We agree with you that the sample size is small. All final year dental students (52 students) were invited to participate, and questionnaires that were partially completed were excluded, leaving a total of 42 questionnaires for analysis.

We could have planned a multicenter study, which would have allowed us to have a larger sample size and to evaluate differences depending on differential factors in the teaching of the different centers, but this was not our objective.

We considered that given the sample size, it was not appropriate to introduce more complex statistical analyses. However, we are open to doing so, if you suggest what kind of more complex analysis could be carried out, given the specific characteristics of the study.

Round 2

Reviewer 2 Report

The authors have responded to the comments satisfactorily. However, the issue of the validity of responses by the participants still remains subjective.

Author Response

Thank you again for your time and interest to improve the quality of the present manuscript and for acknowledging our efforts to comply with your suggestions. As mentioned in the previous review, the options provided as answers for each of the clinical scenarios were based on published systematic reviews and consensus reports whenever possible, and original articles if this was not the case. With regards to the responses provided by the participants, they answered based on their knowledge on the matter, according to their dental training and learning, which is also evidence-based. Therefore, we believe that the methodology used is suitable to fulfill the objectives of the present study: to evaluate overdiagnosis and overtreatment upon different clinical situations among last-year dental students.

Reviewer 4 Report

As I said in my first review, I consider the topic of this study interesting. But because it assesses the perception and the attitude in some clinical cases just for a small group of students from an university, I consider that it has no high scientific relevance.

So, I appreciate the authors’ hard work, but I consider the study is not suitable for a Q1 ranked journal.   

Author Response

Thank you again for your time and interest to improve the quality of the present manuscript and for acknowledging our efforts to comply with your suggestions. The objective of the present study was to evaluate overdiagnosis and overtreatment upon different clinical situations among last-year dental students. To do so, a sample of students from our university was selected from which 80.77% participated.

Indeed, it would be of interest that dental education programmes would be standardized and unified with regards to the different training criteria. However, it is not the case in Spanish universities. If a multi-centric sample was selected, the objective would have to be different to the one which was proposed.

Additionally, with regards to the scientific relevance of this study, to our knowledge, there is a lack of studies on the assessed topic. For this reason, the results from this study can contribute to fill this gap in the knowledge and are therefore relevant.